# Identification of Genomic Regions for Traits Associated with Flowering in Cassava (*Manihot esculenta* Crantz)

**DOI:** 10.3390/plants13060796

**Published:** 2024-03-12

**Authors:** Julius K. Baguma, Settumba B. Mukasa, Ephraim Nuwamanya, Titus Alicai, Christopher Abu Omongo, Mildred Ochwo-Ssemakula, Alfred Ozimati, Williams Esuma, Michael Kanaabi, Enoch Wembabazi, Yona Baguma, Robert S. Kawuki

**Affiliations:** 1School of Agricultural Sciences, Makerere University, Kampala P.O. Box 7062, Uganda; settumba.mukasa@mak.ac.ug (S.B.M.); nuwamanyaephraim@gmail.com (E.N.); mildred.ochwossemakula@mak.ac.ug (M.O.-S.); 2National Crops Resources Research Institute, Namulonge (NaCRRI), Kampala P.O. Box 7084, Uganda; talicai@hotmail.com (T.A.); caomongo@gmail.com (C.A.O.); ozimatialfred@gmail.com (A.O.); esumawilliams@yahoo.co.uk (W.E.); kanaabimichael@gmail.com (M.K.); earlmbabazi@gmail.com (E.W.); kawukisezirobert@gmail.com (R.S.K.); 3National Agricultural Research Organisation (NARO), Entebbe P.O. Box 295, Uganda; ybaguma@naro.go.ug; 4School of Biological Sciences, Makerere University, Kampala P.O. Box 7062, Uganda

**Keywords:** flowering traits, first branching, branching levels, flowering behavior, functional annotations

## Abstract

Flowering in cassava (*Manihot esculenta* Crantz) is crucial for the generation of botanical seed for breeding. However, genotypes preferred by most farmers are erect and poor at flowering or never flower. To elucidate the genetic basis of flowering, 293 diverse cassava accessions were evaluated for flowering-associated traits at two locations and seasons in Uganda. Genotyping using the Diversity Array Technology Pty Ltd. (DArTseq) platform identified 24,040 single-nucleotide polymorphisms (SNPs) distributed on the 18 cassava chromosomes. Population structure analysis using principal components (PCs) and kinships showed three clusters; the first five PCs accounted for 49.2% of the observed genetic variation. Linkage disequilibrium (LD) estimation averaged 0.32 at a distance of ~2850 kb (kilo base pairs). Polymorphism information content (PIC) and minor allele frequency (MAF) were 0.25 and 0.23, respectively. A genome-wide association study (GWAS) analysis uncovered 53 significant marker–trait associations (MTAs) with flowering-associated traits involving 27 loci. Two loci, SNPs S5_29309724 and S15_11747301, were associated with all the traits. Using five of the 27 SNPs with a Phenotype_Variance_Explained (PVE) ≥ 5%, 44 candidate genes were identified in the peak SNP sites located within 50 kb upstream or downstream, with most associated with branching traits. Eight of the genes, orthologous to Arabidopsis and other plant species, had known functional annotations related to flowering, e.g., eukaryotic translation initiation factor and myb family transcription factor. This study identified genomic regions associated with flowering-associated traits in cassava, and the identified SNPs can be useful in marker-assisted selection to overcome hybridization challenges, like unsynchronized flowering, and candidate gene validation.

## 1. Introduction

Cassava (*Manihot esculenta* Crantz, 2n = 36) is a valuable food crop in fighting hunger and poverty in developing countries [1,2]. Additionally, it is a source of animal feed, as well as a raw material for diverse industries [3]. Because of this economic importance of the crop, much of the ongoing breeding efforts are directed towards developing cassava varieties that target different industrial and food products [4,5]. This, however, is hampered by a number of constraints, including poor flowering and limited knowledge of the genetics and inheritance of flowering traits [5,6]. To meet the growing demand for cassava and to overcome the challenges associated with poor flowering, it is imperative to expand knowledge of the genetics of flowering.

Cassava is preferentially propagated by vegetative means using stem cuttings [7,8] due to the difficulty in obtaining botanical seed and low germination and seedling establishment and survival rates. This often results in small and weak plants with comparatively low root yields [8]. Despite these seed-related challenges, seed remains an indispensable and exceedingly important resource in cassava breeding to generate new breeding lines with variations that may be beneficial [7,8,9]. In conventional cassava breeding, beneficial genetic variations are derived through hybridizations that involve crosses under controlled or open-pollination, resulting in full or half sib progenies [8]. Thus, introgression of economically important traits requires the selection of progenitors with profuse, timely, and synchronized flowering. However, many of the progenitors with desirable farmer preferences are often poor at flowering, while others never flower at all. In addition, there is great disparity in the time and amount of flowering among many cassava genotypes, with some flowering early and profusely, while in others, flowering is late and scarce [5,6,10]. The early flowering or branching types tend to have shorter main stem heights, while the late branching ones have taller stem heights at first branching. All this indicates that cassava flowering is complex, exhibiting varied patterns that make synchronization of crosses difficult [11]. Altogether, these challenges complicate the process of progenitor selection by the breeder, who is obligated to choose genotypes with the capacity to flower but without compromising farmer preferences.

Flowering in cassava begins with morphological changes in plant architecture that involve branching of the main stem [10]. Flower bud formation occurs at branch points on the stem [8,10] as axillary buds below the inflorescence allow upward growth of the plant. This implies that a branching genotype is a flowering type, and every branching event results in flower formation [12]. The successive branching may be di-, tri-, and tetra- or pentachotomous, resulting in several branching events [8]. The highly branching genotypes flower more prolifically than those with sparse branching, while the non-branching (erect) ones are usually non-flowering. Most farmer-preferred cassava genotypes are either erect or non-branching and, coincidentally, are non- or late flowering types, which rarely produce seeds [5,12]. The correlation between branching behavior and flower production observed in cassava suggests that these traits might share a similar genetic basis. However, knowledge of genes controlling this kind of flowering correlation and/or behavior in the crop is scanty.

The flowering process in angiosperms is evoked by an interaction of environmental and endogenous signals [13,14,15,16]. This is known to be regulated by a sophisticated network that monitors changes in the environment to ensure that flowering occurs under suitable conditions for maximization of reproductive success and seed production [17,18]. Like in other plants, studies have indicated that the flowering response in cassava is under the influence of environmental factors, such as the photoperiod and temperature [19,20,21]. The understanding of the control mechanisms of flowering in many plants is mainly based on the model plant *Arabidopsis thaliana*, in which various genes, including *FLOWERING LOCUS T (FT)*, and proteins have been implicated [17]. *FT* homologs such as *MeFT1* and *MeFT2* have been associated with photoperiodic induction and early flowering in cassava [21,22]. Despite this knowledge, the precise mechanism underlying the flowering behavior exhibited in the crop is not clearly known. Thus, deciphering the genetic basis of this flowering behavior and how it is regulated will provide important insights that could be useful in overcoming hybridization challenges, especially flowering synchronization.

Molecular marker-assisted selection (MAS) has been a primary approach in plant breeding for understanding the genetic basis as well as the discovery of functional genomic regions (genes) associated with target phenotypic traits in many plant species [23]. It has been used in the selection of quantitative trait loci (QTL) from breeding populations and introgression of genes from breeding lines or wild relatives [24]. Several QTLs for flowering-related traits have been detected in several crop species, for example, flowering time in rapeseed (*Brassica napus* L.) [25], pistillate character in castor oil (*Ricinus communis* L) [26], and seed traits in jatropha (*Jatropha* spp.) [27]. MAS has been used to detect QTLs or genomic regions for various traits in cassava breeding [23], for example, root yield and starch content [28], plant and first branch height [29], starch biosynthesis [30], and biotic stress resistance [31]. However, knowledge of genes controlling flowering and associated traits in cassava is still limited.

Genome-wide association analysis (GWAS) is an alternative genomic approach that has been designed to overcome the drawbacks of QTL mapping [32]. It is used to identify genetic variants statistically associated with a specific trait of interest using individuals that are ancestrally similar but that differ phenotypically [33,34]. It is robust and thus has been used to identify many true genotype–phenotype associations [35] in several plants, for example, Arabidopsis (*A. thaliana*) [34,36,37], rice (*Oryza sativa*) [38], maize (*Zea mays*) [39,40], rapeseed (*Brassica napus*) [41], and foxtail millet (*Setaria italica*) [42]. GWAS has also been successfully utilized to unravel the genetic architecture of several phenotypic traits in cassava, for example, the starch content [43], dry weight content [44], beta carotene content [44,45,46], disease-resistant traits such as cassava mosaic disease (CMD) [47] and cassava brown streak disease [48,49], and drought tolerance [50].

Relatedly, GWAS has been used to elucidate the genetic basis of flowering traits in several crops, for example, flowering time variability in pearl millet (*Pennisetum glaucum* L.) [51], maize (*Zea mays*) [52], common bean (*Phaseolus vulgaris* L.) [53,54], sesame (*Sesamum indicum* L.) [55], and rapeseed (*B. napus* L.) [56]; and flowering and plant height under different levels of nitrogen in Indian mustard (*Brassica juncea*) [57] and soybean (*Glycine max*) [58]. In cassava, reports on the utilization of GWAS to determine the genetic basis of flowering and related traits are scanty, except the first branch height [59]. This study aimed at identifying genomic regions for traits associated with flowering in cassava.

## 2. Results

### 2.1. Phenotypic Variation, Distribution and Heritability Estimates of Flowering-Associated Traits

Phenotyping was performed for flower (staminate and pistillate) counts and five flowering-associated traits on a total of 293 cycle two (C2) cassava accessions. This was performed for two growing seasons (during 2019 and 2020) and two locations. Branching was either zero (did not branch) or one (branched), branch type or habit ranged from zero to four, branching levels were maximum at 12, while number of nodes ranged from 0 to 148 and stem height at first branching ranged from 0 to 231 cm (Table 1). The coefficient of variation (CV) values varied from 26.87% for branching to 54.66% for nodes.

Broad-sense (*H*^2^) and SNP (genomic) (*h*^2^) heritability were estimated for each trait across season and location (Table 1). The highest broad-sense heritability (*H*^2^) was exhibited by pistillates (0.83), followed by stem height at the 1st branch (0.60), while staminates had the lowest heritability (0.00). SNP-based heritability ranged from the lowest (0.00) for branch type to the highest (0.35) for stem height (at the 1st branch). Negatively skewed (-) distribution trends were observed for branching and branching levels, while branch type, nodes and stem height at first branching, as well as the number of flowers, had positively skewed (+) distributions (Figure 1). Results of the ANOVA demonstrated that the differences among the cassava accessions (*p* ≤ 0.001) used in this study were highly significant for all traits and the magnitude was even higher for the seasons and locations, except for branch type, which was not significant across locations (Table 2).

### 2.2. Correlation between Flowering and Flowering-Associated Traits

Spearman’s correlation coefficient was applied to estimate correlations among flower types and flowering-associated traits considered in this study. All the traits showed significant positive correlations ranging between 0.66 and 1.00 (Appendix A). Generally, relatively lower correlation values were noted between pistillates and other traits associated with flowering.

### 2.3. Genotyping and SNP Identification

Genotyping enabled the identification of 24,040 SNP markers, all of which were mapped onto the 18 cassava chromosomes (Figure 2). Upon filtering to remove SNPs with missing genotypes or markers with a minor allele frequency (MAF) of 1%, 22,103 SNPs were retained. The SNP coverage per chromosome ranged from 806 SNPs on chromosome 16 to 2065 SNPs on chromosome 1. The polymorphism information content (PIC) averaged 0.25 (ranging from 0.02 to 0.38) and MAF averaged 0.23 (ranging from 0.01 to 0.50).

### 2.4. Population Structure and Kinship

The population structure of the 293 accessions was estimated with principal component analysis (PCA) and a kinship matrix to reveal the genetic relationships. Three distinct subpopulations were recognized (Figure 3A), where each colour represents a different genetic cluster. The list of genotypes and the overall representation of membership of the sample in each of the three clusters are presented in Appendix A. In the structure, the clusters consisted of 118, 106 and 69 genotypes in clusters 1, 2 and 3, respectively. Kinship among accessions estimated using a marker-inferred kinship matrix revealed shared genetic relationships between the accessions (Figure 3B). There were familiar relationships along the diagonal with a few large blocks of closely related individuals. The off-diagonal part of the relationship matrix indicated low kinship.

### 2.5. Linkage Disequilibrium Estimation

The decay rate of LD was measured as the chromosomal distance at which the average pairwise correlation coefficient (*r*^2^) among markers dropped to half its maximum value. The LD and LD decay distance in all 18 chromosomes are summarized in Figure 4. The average *r*^2^ value for 18 chromosomes was 0.32 observed at a distance of ~2850 kb (kilobase pairs) and that value slowly dropped to half (0.16) at distance of ~4828 kb. The maximum decay distance was ~32,012 kb on chromosome 7 and 0.009 kb on chromosome 10. The lowest *r*^2^ value was 0.12 with a maximum decay distance of ~39,193 kb on chromosome 8 and a minimum decay distance of 0.006 kb on chromosome 15. Meanwhile, the highest *r*^2^ value was 0.95 with a maximum decay distance of ~796 kb on chromosome 3 and a minimum decay distance of ~33 kb on chromosome 10.

### 2.6. Marker–Trait Association Mapping

A marker–trait association (MTA) analysis using combined data from two locations of 293 cassava accessions and 22,103 SNP markers was undertaken using the mixed linear model (MLM) approach. Manhattan and quantile–quantile (Q-Q) plots describing significant SNP associations are shown in Figure 5. The Manhattan plots of the five flowering-associated traits were based on the combination of >1000 BLUPs. A total of 53 significant MTAs (involving 27 SNPs) were identified as being associated with traits associated with flowering, and these were distributed mainly on chromosome 18, which had 20 of the 53 MTAs (Appendix A). Branching (Branch) had 15 MTAs on chromosomes 5, 8, 11, 15, 16, and 18; branch type at first branching (Branch1_No.) was associated with 8 SNPs on chromosomes 5, 8, 14, 15, and 18. Similarly, the number of branching levels (Branch_Levels) had 14 MTAs on chromosomes 5, 8, 9, 11, 14, 15, and 18, while the number of nodes at first branching (Branch1_Nodes) had the highest associated markers (16) on chromosomes 3, 5, 8, 9, 14, 15, and 18. Meanwhile, stem height at first branching (Branch1_Ht) had no MTAs based on the Bonferroni threshold (Figure 5). The Q-Q plot showed the distribution of observed and expected *p*-values (association test statistics). The Q-Q plot analysis for all traits, except stem height, showed a fairly midway deviation of the observed *p*-values from those expected.

### 2.7. Identification of Candidate Genes

Gene annotation information according to Phytozome v13 (*Manihot esculenta* v7.1) (https://phytozome-next.jgi.doe.gov/) (URL accessed on 23 February 2023) and the National Centre for Biotechnology Information (NCBI) of cassava, *Manihot esculentum*, genome (Manes) (https://blast.ncbi.nlm.nih.gov/) (URL accessed on 23 February 2023) were used to determine the putative functionality of genes around the associated marker loci. Out of the 27 SNPs involved in MTAs, five SNPs with Phenotype_Variance_Explained (PVE) ≥ 5% were selected to predict genes associated with the flowering traits. Two of these SNPs, S15_11747301 (on chromosome 15) and S18_1832353 (on chromosome 18), had associations across the two study locations, while S3_21330906 (on chromosome 3) and S5_22566689 (on chromosome 5) were specific to Serere, and S5_29309724 (on chromosome 5) was specific to Namulonge (Appendix A). A total of 44 candidate genes located within 50 kb upstream or downstream of the five markers in the cassava genome were found to be associated with four (branching, branch type, branching levels, and nodes at first branching) of the five traits associated with flowering (Appendix A). SNPs S5_29309724 and S15_11747301 were associated with all four traits and had 5 and 14 putative genes, respectively. SNP S18_1832353 was associated with two traits involving 12 genes, while the remaining SNPs were associated with one trait each, with varying numbers of putative genes. Detailed information about all significant associations is summarized in Appendix A.

Marker *R*^2^ values ranged from 0.097 to 0.195, with *p*-values varying from 6.39 × 10^−12^ to 5.16 × 10^−7^, while PVE values considered for gene identification ranged from 4.84, attributed to S18_1832353 (associated with Branch_Levels), to 36.88%, attributed to S5_29309724 (associated with Branch1_No.) (Appendix A). Thirteen genes (nine on SNP S3_21330906 and four on S5_22566689) were unique to the nodes at 1st branch, but the remaining 27 genes were associated with all four flowering-associated traits. A total of 19 gene loci were each associated with four traits, for example, *Manes.15G140500* (eukaryotic translation initiation factor 3 subunit I) and *Manes.15G140900* (myb family transcription factor MOF1). Similarly, 12 candidate genes, such as *Manes.18G016700* (aldehyde oxidase GLOX) and *Manes.18G016800* (lysine-specific histone demethylase 1 homolog 3), were associated with two traits.

Candidate genes with known functions and orthologous with Arabidopsis or other plant species were identified using five SNPs with a PVE ≥ 5% (Table 3). Among the genes found to be associated with these SNPs, this report focussed only on those reported to be functionally controlling flowering or flowering-associated responses in various plant species. As such, eight genes were identified as candidates and all were associated with branching, branch type, and branching level traits. For example, *Manes.18G016700* (aldehyde oxidase GLOX) as well as *Manes.18G016725* (lysine-specific histone demethylase 1 homolog 3) and *Manes.18G016200* (cytochrome P450 83B1), all of which are associated with SNP S18_1832353 on chromosome 18, are involved in anther and pollen development [60], promotion of the floral transition [61], auxin homeostasis, and reproductive development [62,63], respectively in *A. thaliana*. *Manes.05G186700* (protein DETOXIFICATION 48) associated with SNP S5_29309724 on chromosome 5 is involved in specifying the lateral organ initiation rate in *A. thaliana* [64]. Associated with SNP S15_11747301 on chromosome 15 is *Manes.15G140500* (eukaryotic translation initiation factor 3 subunit I), which was reported to regulate negatively translation during flower development [65]. *Manes.15G140900* (myb family transcription factor MOF1) plays a role in the regulation of organ identity and spikelet meristem determinacy in rice [66], and *Manes.15G140300* (non-specific lipid-transfer protein 4.1) is involved in seed and ovule maturation and development in *Hordeum vulgare* (barley) and *A. thaliana* [67]. *Manes.15G140600* (short-chain dehydrogenase reductase ATA1), also located on chromosome 15, was reported to play a role in tapetum development in *A. thaliana* [68]. The new genes identified in the present study are promising candidates for follow-up studies on the validation of genes controlling flowering and associated traits in cassava.

## 3. Discussion

In this study, GWAS was used to detect genomic regions associated with the flowering process in a diverse population of 293 cassava accessions. This enabled the generation of knowledge to enhance our understanding of the genetics of traits associated with cassava flowering, i.e., branching, branch type, branching levels and number of nodes of the main stem at first branching.

### 3.1. Variation in Phenotypic Traits Related to Flowering

The ability to flower, as well as early and synchronized flowering, are important target traits for the selection of progenitors in cassava hybridization programmes, but this does not have to compromise traits preferred by farmers. Branching, branch type, branching levels, number of nodes, and stem height at first branch are major indicators of flowering ability in cassava. Commonly, it is through flowering and pollination that crucial genetic variation is generated, which is fundamental for the success of plant breeding programs [72]. Understanding phenological variabilities exhibited in cassava flowering is essential for the accurate selection of parents and planning crosses in hybridization programs. However, these different phenologies are difficult to understand due to the complex genetics of the crop.

The number of nodes followed by stem height exhibited the highest variability. There is greater potential for the selection of progenitors with the capacity to flower basing on these attributes compared to others. High and exploitable variability in flower and fruit numbers, as well as the time to flower, in cassava was also reported in a previous study [11]. Conversely, the lowest variation recorded for branching indicates low exploitable variability compared to other traits.

Broad-sense heritability varied across traits, i.e., pistillates had the highest heritability (*H*^2^ = 0.83), followed by stem height at the first branch (*H*^2^ = 0.60). There are not many reports on broad-sense heritability estimates for flowering and associated traits in cassava; however, heritability estimates of over 0.80 were reported for plant height at first branch, number of branches, branching levels, as well as female flowers [11,73,74,75]. SNP-based heritability (*h*^2^) estimates were relatively higher for stem height at the first branch (*h*^2^ = 0.35), followed by branching (*h*^2^ = 0.25), and branching levels (*h*^2^ = 0.19). These estimates could be attributed to LD unevenness or heterogeneity among marker regions due to overestimation or underestimation, as postulated by [76].

### 3.2. Correlation between Flowering and Flowering-Associated Traits

A correlation coefficient analysis is important for measuring the degree and direction of relationships between various traits. The positive and significant correlations among the traits in this study, such as branch and branch type (*r*^2^ = 0.95) or flowers (*r*^2^ = 0.94), branching levels and pistillate flowers (*r*^2^ = 0.76), branching levels and height of first branch (*r*^2^ = 0.88), etc., was consistent with the previous findings by [77]. Thus, this reflects a genetic relationship between the traits, and selection for one trait would directly affect the expression of the other trait, hence facilitating the selection of progenitors to include in a breeding program. Relatedly, a correlation between branching levels and height of the first branch was reported in a previous study on the induction of cassava flowering [78]. Overall, all these traits are dependable indicators of the flowering response, suggesting genetic linkedness.

### 3.3. Population Structure and Kinship

Population stratification and the degree of kinship of the samples are critical factors in genetic studies including GWAS [79,80]. Whereas population structures show the presence of distinct genetic subgroups within a population [81], kinships provide valuable insights into the degree of genetic relatedness between the individuals within a population [79]. PCA and kinship analysis revealed clustering of the germplasm into subgroups, including three major subgroups, according to the kinship heatmap. The clusters (within the PCA and kinship plots) show genetic relatedness within members of the subgroup, suggesting a common ancestry within the subgroup. Clusters along the diagonal line in the kinship plot show unrelatedness among groups, while pairs of individuals along the diagonal line show individuals’ self-kinship. Shorter connecting lines between individuals or clusters indicate closer relationships while longer connecting lines indicate more distant relationships.

The Q-Q-plot analysis further revealed subgroups in the study population. Plots for all traits, except stem height, showed a fairly midway deviation of the observed *p*-values from those expected, towards the top left. This indicated that, firstly, the SNPs associated with these traits were significant and, secondly, the existence of genetic variations between the subpopulations. Thus, the results suggested that genetic variants have different allele frequencies among subgroups of the population investigated. For stem height at first branching, the little or non-segregation between the observed and expected *p*-values suggests that the sub populations are genetically homogeneous, indicating a similarity in the distribution of allele frequencies between the subpopulations.

### 3.4. Linkage Disequilibrium

LD, a non-random association or correlation between alleles at different genetic loci within a population, plays an integral role in genetic studies, such as GWAS and genomic selection [82,83]. When LD exists between a marker (e.g., a SNP) and a phenotypic trait, it implies that the marker and the causative variant of that trait are physically close to each other on the same chromosome [83].

There is limited knowledge about the genetic architecture and potential markers specifically associated with flowering traits in cassava because little research has been conducted on the LD patterns and LD decay for these traits. In this study, the average LD among the SNP marker pairs of 293 cassava accessions was 0.32 at a distance of ~2850 kb, which slowly declined to 0.16 at distance ~4828 kb, where it became more or less stable. A study by [84] on the evaluation of genetic diversity among 1580 cassava accessions presented a lower LD value average of 0.014, though the initial LD decline was more rapid before it stabilized at 15 to 20 kb. Additionally, an analysis of a panel of 876 cassava accessions belonging to the Cassava Germplasm Bank of Embrapa reported LD of *r*^2^ < 0.1 that declined to between 0.3 and 2.0 Mb [85]. Meanwhile, whole-genome LD decay peaked at *r*^2^ of 0.349 and dropped to an *r*^2^ of 0.212 at a distance of 10 kb in a study on the genetic architecture of defensive, agro-morphological and quality-related traits involving a panel of 5130 cassava clones [86]. These observations show that the LD estimates and decay rates in the cassava genome are low and inconsistent.

The low and slow decline in LD observed in this study and the observed inconsistencies among the reported LD values of the different studies could be attributed to the genetic complexity (high heterozygosity) of the crop as well as the vegetative mode by which the crop is predominantly propagated. Additionally, the artificial selection pressures imposed by preferential selections based on specific traits, like disease resistance and high yield, tends to favour specific alleles linked to the desired traits, leading to extended LD around these beneficial variants. In potato (*Solanum tuberosum* L.), a clonally propagated crop but outcrossing and highly heterozygous like cassava, LD decay was reported to be relatively fast in the short range (*r*^2^ = 0.208 at 1 kb) but slowed afterward (*r*^2^ = 0.137 at 70 kb) [87]. Perennial or clonally propagated plants, including cassava, generally have long breeding cycles and thus show a limited number of recombination cycles. Hence, their LD decays are relatively slow [88] in spite of the outcrossing nature of these crops.

### 3.5. Marker–Trait Association Mapping

Association or LD mapping, also known as association analysis or GWAS, is an important tool in plant breeding for understanding the genetic basis of complex traits. It involves surveying genetic variations in the whole genome to find signals of associations between genetic markers, such as SNPs, and various phenotypic traits of interest within a diverse population [82]. It is also used in identifying candidate genes and genomic regions that are presumed to be associated with traits of interest [82,89].

All traits, except stem height at first branching, had MTAs on different chromosomes, suggesting that these are complex traits under the influence of multiple genes. For example, branching was associated with 15 markers on chromosomes 5, 8, 11, 15, 16, and 18, while branching levels had 14 MTAs on chromosomes 5, 8, 9, 11, 14, 15, and 18. This shows that these traits are undoubtedly polygenic in nature. Meanwhile, some chromosomes had more than one MTA for some traits, for example, chromosome 18 was associated with eight, four, six and three MTAs for branching, branch type, branching levels and number of nodes at first branching, respectively. The high phenotypic trait correlations might explain this pleiotropic effect. This also suggests that these traits can be coinherited. This can be advantageous because it would enable narrowing the search for traits of interest. A study by [85] also revealed multiple MTAs on chromosome 18 for cassava starch pasting properties, indicating the high LD of markers on this chromosome. Multiple MTAs have also been detected for flowering traits in Indian mustard (*Brassica juncea*) [57] and grain yield and related traits in durum wheat (*Triticum turgidum*) [90]. Other traits had only one MTA on a chromosome, such as S15_11747301 on chromosome 15 (associated with branching, branch type, branching levels, and number of nodes). This suggests that the marker may be associated with a potential gene or genomic region primarily responsible for influencing the given traits on that particular chromosome. This may be particularly helpful in genome editing.

The absence of MTAs for the trait of stem height at first branching, in spite of a higher SNP heritability (*h*^2^ = 0.35), could be attributed to the genetic complexity or polygenic nature of the trait, involving multiple genes with small effects. In cassava, like most crops, it has been shown that the height and branch height traits are controlled by multiple genes [91]. In polygenic traits, individual markers do not show significant associations with phenotypic traits because the genetic contribution is due to an additive effect distributed across many loci, each with a small impact [92,93]. This probably was the cause of a failure to detect MTAs for height in this study. On the other hand, branch type and the number of nodes had very low SNP heritabilities but had significant MTAs. In a related study, though not comparable, significant MTAs were detected for fruit characteristics, including those with very low heritability estimates, in interspecific pear (*Pyrus*) [94]. The low heritability estimates of the traits in this study, yet with significant MTAs, may have been due to environmental effects or genetic heterogeneity of the population used in this study. Together, these results demonstrate that the flowering-associated traits in cassava have a genetic basis, and they may be the result of shared underlying genetic mechanisms controlling the traits.

### 3.6. Putative Candidate Genes Linked to Marker Loci for Flowering-Associated Traits

In cassava, branching of the main stem is associated with the transition from vegetative growth to reproductive development. This indicates the onset of flowering with flower bud formation being preceded by a highly genetically planned apical branching [8], and the genes identified in this study could be orthologous with those that control the mechanism underlying the initiation of branching and the floral transition.

In this study, branching, branch type, and branching levels had commonality in 27 of the 37 predicted genes. Of these, eight were reported to be functionally annotated for processes directly or indirectly leading to flowering responses in different plant species. For example, *Manes.18G016725* (lysine-specific histone demethylase 1 homolog 3), associated with SNP S18_1832353 on chromosome 18, was reported to influence the transition from vegetative growth to the flowering phase in *A. thaliana* [61]. It reduces levels of histone methylation in chromatin of key flowering genes such as the floral repressor *FLOWERING LOCUS C(FLC)*, which must be preferentially expressed in shoot and root apical regions during a plant’s vegetative development. This enables the achievement of precise levels of *FLC* required for the onset of a flowering response.

*Manes.18G016200* (cytochrome P450 83B1) located in close proximity with SNP S18_1832353 on chromosome 18 is involved in auxin and indole glucosinolate biosynthesis in Arabidopsis [62]. Auxins are plant growth regulators integrated within the gene regulatory networks that control most aspects of plant development, including coordinated organ outgrowth and initiation of floral meristems [95]. Additionally, cytochrome P450 proteins, which are widely distributed in plants, participate in a vast array of pathways leading to the synthesis and modification of multiple metabolites with variable and important functions during different stages of plant development [63]. Meanwhile, although the role of *Manes.18G016700* (aldehyde oxidase GLOX, Germin-like oxalate oxidases) in plants is not yet fully understood, it is believed to be involved in anther development, during which it plays a role in tapetum and pollen development in *A. thaliana* [60]. It is also involved in the catalyzation of the oxidation of aldehydes to the corresponding carboxylic acids [69], including the oxidization of indole acetaldehyde into indoleacetic acid (IAA), the most important form of auxin [96].

*Manes.05G186700* (protein DETOXIFICATION 48), which is associated with SNP S5_29309724 on chromosome 5, a member of multidrug and toxic compound extrusion (*MATE*) gene family, is involved in specifying the lateral organ initiation rate in *A. thaliana* [64]. It has been implicated in axillary bud formation in the shoot apical meristems, resulting in accelerated organ formation, including leaf formation and early flowering. It is also involved in detoxification or extrusion of toxins and regulation of plant development, including fruit development [97]. *Manes.15G140500* (eukaryotic translation initiation factor 3 subunit I) is associated with SNP S15_11747301 on chromosome 15 and is a member of eukaryotic translation initiation factor 3 (eIF3) complex [65]. It was reported to regulate negatively translation during flower development in *A. thaliana*. It is responsible for initiating protein synthesis during development, the germination of pollen grains, and embryogenesis. Its expression was detected in plant cells and tissues differentiating into organs, including inflorescences, implying its involvement in influencing the transition from vegetative to reproductive growth [65]. *Manes.15G140900* (myb family transcription factor MOF1), also associated with SNP S15_11747301, plays a role in the regulation of organ identity and spikelet meristem determinacy in rice [66]. It causes a delayed transition from the spikelet to the floral meristem, which may result in defective or sterile spikelets.

*Manes.15G140300* (non-specific lipid-transfer protein 4.1), which is associated with S15_11747301 on chromosome 15, is a protein reported to be involved in numerous biological processes, such as the transfer of phospholipids and reproductive development in plants, including maize [71,98]. It also plays a role in seed and ovule maturation and development, probably by regulating the fatty acids homeostasis during suberin and sporopollenin biosynthesis or deposition [98]. Additionally, *Manes.15G140600* (short-chain dehydrogenase reductase ATA1), also near SNP S15_11747301, was reported to play a role in tapetum development in *Silene latifolia* and *A. thaliana* [68].

## 4. Materials and Methods

### 4.1. Plant Material and Field Trials

A diverse panel consisting of 446 cassava accessions with unknown and/or diverse flowering behaviors were used in this study. These accessions were derived from a cycle two (C2) population that was developed at the National Crops Resources Research Institute (NaCRRI). Briefly, the C2 population resulted from successive cycles of selection and hybridization [99,100]. These accessions were established in trials laid out in an augmented design with three checks, namely, Mkumba, TME 204, and NAROCASS 1. The clones Mkumba and NAROCASS 1 are known for early forking and/or profuse flowering, while TME 204 is a late or non-forking clone. Each plot had one row of 10 plants spaced at 1 m between plants and rows. The trials were established at two locations: (1) Namulonge in Central Uganda (0.52166458° N, 32.608997564° E) at an altitude elevation of about 1150 m above sea level (asl), with a natural photoperiod of about 12 h, which is fairly uniform throughout the year. It is characterized by an average annual rainfall of approximately 1300 mm, average annual temperature of 22 °C, and annual minimum and maximum temperature of 16 and 28 °C, respectively. (2) National Semi Arid Resources Research Institute (NaSARRI), Serere located in Eastern Uganda (1.4994° N, 33.5490° E) at 1140 m asl with a 12 h photoperiod, low annual rainfall of 900–1300 mm and annual average temperature of 26 °C. The trials were conducted for two growing seasons, 2019/2020 and 2020/2021, from September to August of each season. All trials were managed according to the institute’s standard agronomic practices for cassava.

### 4.2. Flowering Traits Evaluated

Plant-based counts of flowers (staminate and pistillate) were commenced at about three months after planting (MAP), and this was continued every seven days. However, this collection was conducted only for one season due to the interruption by the COVID-19 lockdown in 2019 and 2020. Meanwhile, data on the five traits associated with flowering were collected on a per plant basis at 12 MAP. The traits included (i) first branching of the main stem (Branch), which was scored as “1” for branched or “0” for non-branched; (ii) branching type or habit (Branch1_No.) as the number of branches at the first branching event or tier 1; (iii) number of branching levels of the main stem (Branch_Levels); (iv) number of nodes (Branch1_Nodes) counted with tally counters up to the first branch; and (v) stem height to first branching (Branch1_Ht) from the ground to the first branch of the main stem, measured with a meter ruler or tape measure. These traits were prioritized based on reports that inferred that stem branching was an indication and precondition for flowering in cassava [7,8,10]. Data on these traits were collected for two consecutive seasons from 2019–2021 in two locations.

### 4.3. Genomic DNA Extraction and Genotyping

Extraction of genomic DNA and genotyping were carried out as described in [48]. Briefly, two young top leaves were collected from the genotypes, folded, and punched using a 5 mm hand puncher for genomic DNA extraction and purification. Extraction was performed using a DNA extraction protocol routinely used at Intertek-AgriTech (http://www.intertek.com/agriculture/agritech/) (URL accessed on 2 November 2022) and the extracted DNA samples were purified and quality as well as quantity were evaluated before being genotyped. Genotyping was performed using the Diversity Array Technology Pty Ltd. (Canberra, Australia) (DArTseq) genotyping platform (https://www.diversityarrays.com/technology-and-resources/dartreseq/, accessed on 2 November 2022).

### 4.4. SNP Calling and Annotation

The sequences of the genomic representations were aligned to cassava reference genome v6.1, resulting in the selection of 28,434 raw SNP markers. The genotype data were then subjected to stringent filtration measures to remove genotypes with >10% and SNPs with >5% missing data or with a minor allele frequency of less than 5%. Thus, a total of 24,040 high quality SNP markers were used for GWAS. The SNPs were annotated on the basis of the cassava reference genome v6.1. The markers were converted to the dosage format of 1, 0, 2, which represented alternative allele homozygotes, heterozygotes, and reference allele homozygotes, respectively.

### 4.5. Statistical Analyses

All phenotypic data were subjected to various statistical analyses using R Statistical Software v4.2.2 [101]. Accessions that did not have a complete set of data points across seasons and locations were excluded from the statistical analyses, thus leaving 293 out of the 446 initial accessions. Given that this study was conducted in two locations and two seasons, best linear unbiased predictions (BLUPs) for each trait in each accession were computed and used for statistical and GWAS analyses. In this case, an R script based on a linear model described by a previous study [102] was used and the obtained values were used as phenotypes for the association analysis. Stem branching was scored as categorical data (scored as “1” for branched or “0” for non-branched) and thus was analyzed using a generalized linear model (glm) by applying a logistic regression model. The number of branches, branching levels and nodes were analyzed, as recommended for count data, using a negative binomial distribution [103,104], while analysis of variance (ANOVA) was performed for the parameter of stem height. Generalized linear mixed models and *lme4* packages for R statistical software [101] were used. Broad-sense heritability (*H*^2^) per trait was computed using accession and location variance components extracted from the ANOVA models as:$$H^{2}=\frac{V_{G}}{\left(V_{G}+\frac{V_{G}\times E_{G\times L}}{L}+\frac{V_{R}}{L\times S}\right)}$$
where $V_{G}$ was the accession variance; $V_{G}\times E_{G\times L}$ was the variance attributed to the accession by location interaction; $V_{R}$ was the model residual variance; L was number of locations; and S was the number of seasons.

SNP-based heritability (*h*^2^) was computed as:$$h^{2}=\frac{\sigma_{a}^{2}}{\left(\sigma_{a}^{2}+\sigma_{e}^{2}\right)},$$
where $\sigma_{a}^{2}$ was the additive genetic variance and $\sigma_{e}^{2}$ was the residual variance.

The correlation analysis for flowers and flowering-associated traits was conducted based on phenotypic BLUPs. The analyses were performed using the *cor* function, and visualization of the correlation matrices was conducted using the ‘*corrplot*’ package in R [101].

### 4.6. Population Structure and Kinship

A population structure analysis was performed to determine the genetic relatedness between individuals using 24,040 polymorphic SNP markers distributed over the whole genome. Using the Genome Association and Integrated Prediction Tool (GAPIT) 3.0 in R [101,105], we computed a genomic relationship matrix (*K*) (kinship) from the SNP data. Principal component analysis (PCA) of all markers was performed on the genetic relationship matrix using the *prcomp* function in R [101]. The first three principal components (PC1, PC2, and PC3) values were plotted to visualize the population structure.

### 4.7. Linkage Disequilibrium Estimation

A linkage disequilibrium (LD) (in terms of *r*^2^) analysis was performed for the whole genome, as well as individually for each of the 18 cassava chromosomes using package *LDcorSV* in R [101]. To estimate LD decay, the mean *r*^2^ value of each chromosome was computed and plotted against its physical distance to obtain a non-linear regression curve. The LD decay distance was estimated as the physical distance between SNPs where the average *r*^2^ is reduced to half of the maximum LD value [83].

### 4.8. Marker–Trait Association Mapping

The search for MTAs between genotypic and phenotypic data was performed for 22,103 SNPs and the BLUP values obtained for the five flowering-associated traits among the 293 filtered accessions. This was performed using the Genomic Association and Prediction Integrated Tool (GAPIT) in R [101], which determined the mixed linear model (MLM) as the best model at controlling type I errors (false positives) for all traits evaluated. The first five PCs and marker-estimated kinship matrix in each case were used to detect the MTAs. The genome-wide significance thresholds for each association study were assigned using a Bonferroni correction set as 0.1. The significance of associations between traits and markers was assessed on the basis of *p*-values corrected for multiple testing. Manhattan plots, as well as quantile–quantile (Q-Q) plots for association mapping, were used to examine the validity of the SNPs and were visualized using the *qqman* package in R [101].

### 4.9. Identification of Putative Candidate Genes

To identify putative candidate genes residing in close vicinity to high confidence SNPs, the associated SNPs with PVE ≥ 5% were mapped to the reference cassava genome, *Manihot esculenta*, v7.1 using the Phytozome tool v13 (https://phytozome-next.jgi.doe.gov/jbrowse/index.html?data=genomes/Mesculenta_v7_1) (URL accessed on 23 February 2023). Transcripts present within 50 Kb regions from both sides of associated SNPs were fetched along with their description. Identified genes were characterized for gene ontology, including molecular and biological functions, using the UniProtKB tool (https://www.uniprot.org/) (UniProt release 2023_05). Additionally, gene and protein functions were searched using the Alliance of Genomes Resources [106].

## 5. Conclusions

The present study was undertaken to determine the genomic regions associated with traits associated with flowering in cassava under field conditions. The interest in unravelling the genetic basis of cassava flowering is based on the premise that many breeding programs still find it difficult to synchronize flowering time among different cassava genotypes. To date, technologies have been developed to enhance and/or induce flowering in cassava [107,108], but the genotype-dependent responsiveness still makes flower synchronization a significant impediment to cassava breeding. In this study, 53 significant MTAs distributed mainly on chromosome 18 were identified as being associated with flowering behavior in cassava. Five SNPs, i.e., S3_21330906, S5_22566689, S5_29309724, S15_11747301, and S18_1832353 on chromosomes 3, 5, 15, and 18, respectively, were found to be significantly associated with number of nodes and branching traits. Eight genes, i.e., *Manes.18G016700* (aldehyde oxidase GLOX); *Manes.18G016725*; (lysine-specific histone demethylase 1 homolog 3); *Manes.18G016800* (cytochrome P450 83B1); *Manes.05G186700* (protein DETOXIFICATION 48); *Manes.15G140500* (eukaryotic translation initiation factor 3 subunit I); *Manes.15G140900* (myb family transcription factor MOF1); *Manes.15G140300* (non-specific lipid-transfer protein 4.1); and *Manes.15G140600* (short-chain dehydrogenase reductase ATA1) were predicted to be involved in controlling flowering-associated traits, branching, branch type, and branching level traits. The findings of this study, therefore, increase our understanding of the genetics of the flowering process in cassava. However, further studies are required to validate the significant markers and to verify the functions of identified candidate genes using a larger population. Once the functions of the candidate genes have been confirmed, they can be specifically used in marker-assisted breeding, gene editing, and flowering trait introgression to improve flowering, a requirement to develop cassava cultivars with desired traits for food and industry.

## Figures and Tables

**Figure 1 plants-13-00796-f001:**
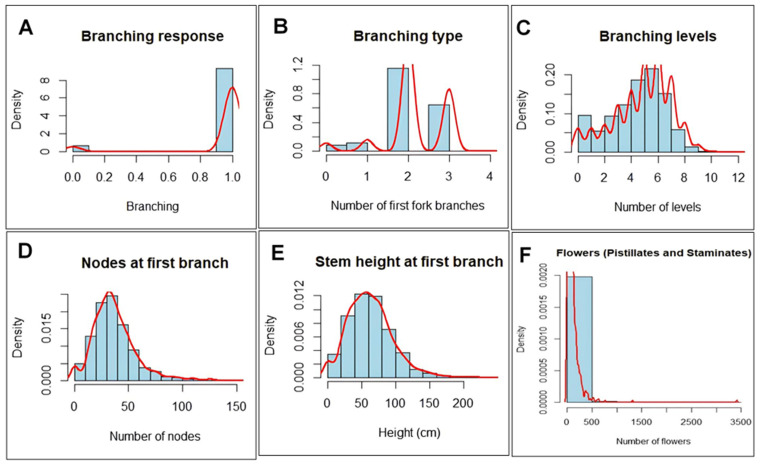
Density distribution of variations in the number of flowers (pistillates and staminates) and flowering-associated traits in cassava.

**Figure 2 plants-13-00796-f002:**
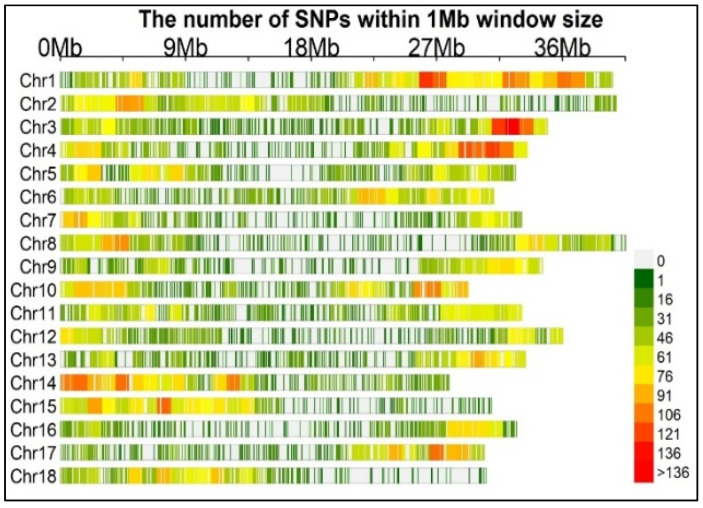
Distribution of SNP markers across the 18 chromosomes. The inserted bar graph represents the number of SNPs (24,040) within a 1 megabase window, and the horizontal axis displays the chromosome length.

**Figure 3 plants-13-00796-f003:**
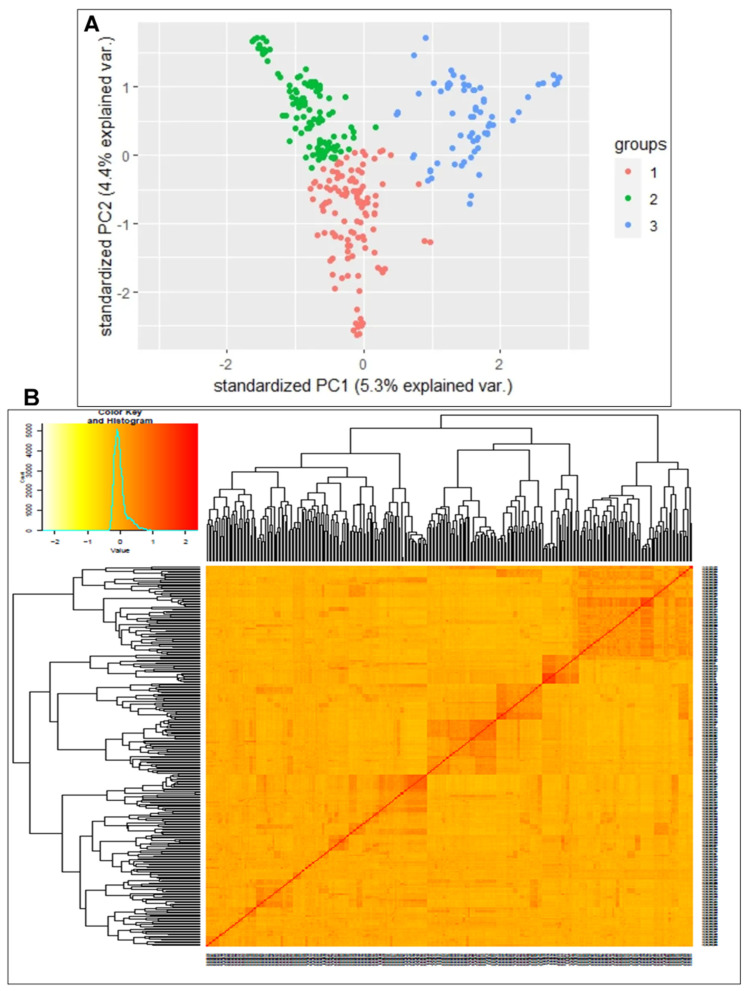
Plots of principal components of 22,103 SNP marker data (MAF > 1%) for 293 individual cassava accessions and kinship. (**A**) displays the scatter plot of PC1 and PC2; (**B**) heatmap showing pairwise genomic relationship matrix, from no relationship (yellow) to a high relationship (red).

**Figure 4 plants-13-00796-f004:**
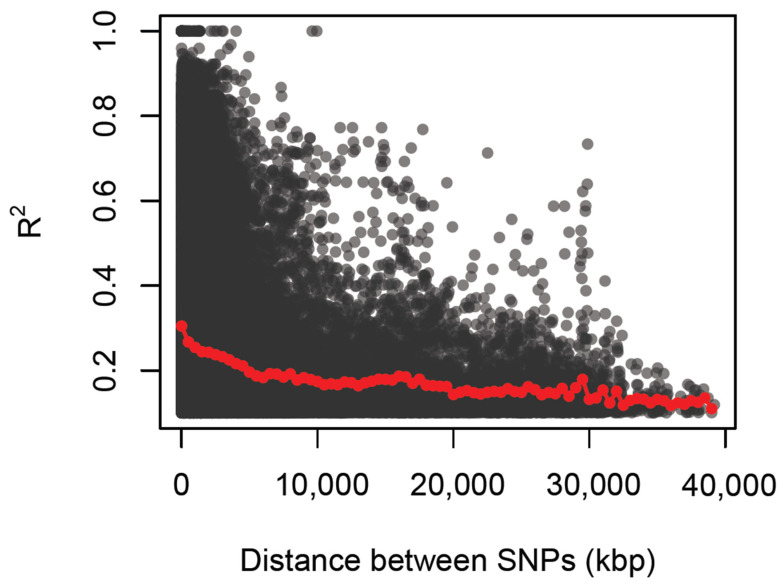
Plot of LD decay of 22,103 SNPs on 18 cassava chromosomes. The red line shows LD decay with physical distance of markers across the genome.

**Figure 5 plants-13-00796-f005:**
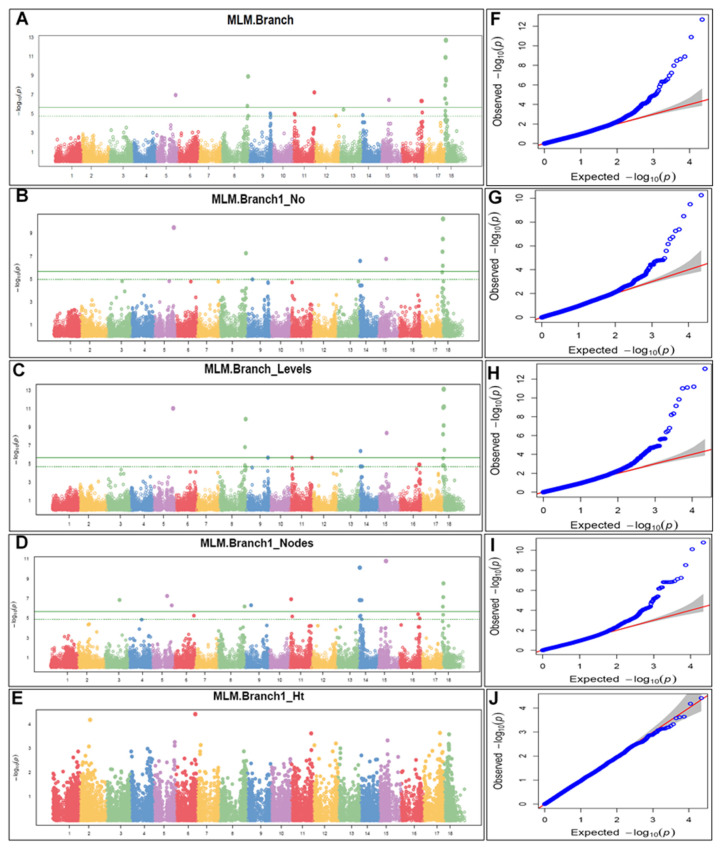
GWAS of five flowering-associated traits across 18 chromosomes in 293 cassava accessions. (**A**–**E**) Manhattan plots of chromosomal positions; (**F**–**J**) quantile–quantile (Q-Q) plots. In all Manihattan plots, the green solid horizontal line indicates a significant line denoting the −log10(P) value significance threshold and the green dashed line indicates a suggestive Bonferroni threshold (−log10(P) = 0.1).

**Table 1 plants-13-00796-t001:** Phenotype variation and heritability estimates of flowers and traits associated with flowering assessed in a panel of 293 cassava accessions.

Variables	Branching	Branch Type	Branching Levels (Number)	Nodes (Number)	Stem Height (cm)	Pistillates (Number)	Staminates (Number)
Observations (n)	1565	1565	1565	1547	1562	801	801
Mean	0.93	2.19	4.86	36.90	63.06	2.09	34.63
Skewedness	-	+	-	+	+	+	+
SEM	0.01	0.02	0.05	0.51	0.85	0.30	8.51
CI (0.95)	0.01	0.04	0.11	1.01	1.67	0.59	16.72
Variance	0.06	0.51	4.60	407.04	1119.56	39.00	31,506.04
SD	0.25	0.71	2.15	20.18	33.46	6.25	177.50
CV (%)	26.87	32.72	44.16	54.66	53.08	2.98	5.13
Heritability, *H*^2^	0.34	0.53	0.38	0.42	0.60	0.83	0.00
SNP heritability, *h*^2^	0.25	0.00	0.19	0.01	0.35	-	-
Significance	*	***	***	***	*	*	***

SEM, standard error of means; CI, confidence interval; SD, standard deviation; CV, coefficient of variation; SNP, single nucleotide polymorphism; -, negatively skewed; +, positively skewed. Tests of significant main effects are indicated: *p* ≤ 0.05 (*) and *p* ≤ 0.001 (***).

**Table 2 plants-13-00796-t002:** Analysis of variance for five flowering-associated traits assessed in 293 cassava accessions.

Variable	Sums of Squares		Mean Squares		*F* Value		Significance	
	Acc	Loc	Acc	Loc	Acc	Loc	Acc	Loc
Branching	57.143	0.634	0.12841	0.63371	3.533	17.4353	***	***
Branch type	435.27	0.96	0.9781	0.9603	3.2537	3.1945	***	ns
Branching levels	5021.7	367.3	11.28	367.26	9.733	316.7597	***	***
Nodes (at 1st branch)	280,458	16954	630.2	16,954	2.4511	65.9356	***	***
Stem height (1st branch) (cm)	755,806	4734	1698	4734	2.2902	6.3829	***	*

Acc, accession; Loc, location. ANOVA tests of significant main effects are indicated: *p* ≤ 0.05 (*), *p* ≤ 0.001 (***), and ns not significant.

**Table 3 plants-13-00796-t003:** Functional annotation of putative genes for flowering-associated traits in cassava.

Trait	SNP	Ch	Gene ID	Description	Function	Reference Species	References
Branch, Branch1_No., Branch_Levels	S18_1832353	18	*Manes.18G016700*	aldehyde oxidase GLOX	Catalyzes the oxidation of aldehydes to the corresponding carboxylates; involved in anther development and plays a role in tapetum and pollen development	*A. thaliana* *Vitis pseudoreticulata* (Chinese wild grapevine)	[60,69]
			*Manes.18G016725*	lysine-specific histone demethylase 1 homolog 3	Promotion of the floral transition	*A. thaliana*	[61]
			*Manes.18G016200*	cytochrome P450 83B1	Functions in auxin homeostasis and plant reproductive development	*A. thaliana*	[62,63]
	S5_29309724	5	*Manes.05G186700*	protein DETOXIFICATION 48	Could be involved in specifying the lateral organ initiation rate	*A. thaliana*	[64,70]
	S15_11747301	15	*Manes.15G140500*	eukaryotic translation initiation factor 3 subunit I	Negatively regulates translation during flower development	*A. thaliana*	[65]
			*Manes.15G140900*	myb family transcription factor MOF1	Transcriptional repressor that plays a role in the regulation of organ identity and spikelet meristem determinacy	*Oryza sativa* subsp. *japonica* (rice)	[66]
			*Manes.15G140300*	non-specific lipid-transfer protein 4.1	Lipid transfer protein involved in seed and ovule maturation and development, probably by regulating fatty acid homeostasis during suberin and sporopollenin biosynthesis or deposition	*Hordeum vulgare* (barley); *A. thaliana*; maize	[67,71]
			*Manes.15G140600*	short-chain dehydrogenase reductase ATA1	May play a role in tapetum development	*A. thaliana*	[68]

## Data Availability

Datasets generated for this study can be found in the article, Appendix A, and at https://cassavabase.org/ftp/manuscripts/Baguma_et_al_2023/ (29 November 2023).

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
