# Peer review of "Identification of Genomic Regions for Traits Associated with Flowering in Cassava (Manihot esculenta Crantz)"

_plants, 2024, doi:10.3390/plants13060796_

Round 1

Reviewer 1 Report

Comments and Suggestions for Authors

An interesting study tackling an important research topic for casava breeding. A few points to consider:

- Serious editing is necessary. A revised copy is attached for reference.

- An augmented design was used to estimate the BLUP. But not much was said to guide the reader on how the analysis was conducted.

- A generalized linear model was used to analyze the categorical traits. No mention of the value considered for residual variance was made which is pre-determined in this context to equal 1 for binary traits or 3.29 on a probit scale. Need to be more specific!

- Some terms were used but not introduced in Mat&Meth. For ex. de-regressed Blups (lines 283-284): something needs to be said on how they were derived. Imputed SNP: imputation implies missing SNP were predicted to fill the blanks and no indication in Mat&Meth that occurred. If not, do not use the term imputed! The traits pistillate and staminate were introduced in correlation analyses without prior explanation on how they were determined.

- Give references for software used in Mat&Meth! Some are missing!

- Some inconsistencies were noted. For ex,  Number of SNP involved in associations with traits: see lines 284-286 where 27 were given once and then 53. Number of SNP retained for analyses and mapping after filtering: 22K and 24k were interchanged in different places, e.g. lines 184-186, 207, 642. Double verify! 

- I do not think Table 4 is really needed when Table 5 gives a more detailed overview.

- The footnote of Fig 6 indicates that the red lines in the plots "compare effects of different models: there is no indication in Mat&Meth which models were considered. I am guessing it is from the GAPIT analyses. So specify!

- It was not clear how the traits considered can help solve the flowering issues in cassava. Are the key traits to fill the gap? What about time to flowering? I was expecting some comparisons between the pistillate and staminate groups in terms of SNP presence and absence and QTL. At least this study would help in  identifying SNP that determine gender, an important aspect of any breeding program. You may need to consider including non-flowering genotypes (if they exist) in the GWAS to better refine the analysis and identify the SNP-QTL that would discriminate these 2 groups (very important for casava breeding). By the way, does applying photoperiod treatments help induce or retard flowering in cassava?

- The Discussion should consider how MAS can make use of these traits and QTL to help a breeding program. Should we expect genotypes that are shy of flowering or do not flower not carrying these SNP? Or there should be something else to induce flowering? The Conclusion should be more forward looking than just a repeat of the results. 

- A revised copy of the manuscript is included to help with the editing needed to improve the manuscript. Comments and suggestions are also provided.

Comments on the Quality of English Language

Avoid being wordy! Use words carefully!

Acronyms are there to simplify the reading. Use them instead of incessantly repeating them with their long meanings.

No need to repeat/list the flowering traits considered in the study time after time.

Reviewer 2 Report

Comments and Suggestions for Authors

The authors identified the genome region associated with flowering traits by GWAS in cassava. As the authors mentioned, cassava flowering is cultivar-dependent, and the elucidation of molecular mechanism of flowering and SNP identification are an important research to overcome the hybridization challenges. In this paper, authors used DArTseq to identify SNPs in genomic DNA extracted from 293 cassava varieties and clarified their relationship with Flowering Trait. This report has an important implications for agricultural science. It's very well written. However, before this report can be published in a journal such as Planta, minor shortcomings need to be addressed. 

As shown in Figure 2, there is certainly a high positive correlation between the number of flowers and branching characteristics, but is this a bipolar distribution? As shown in the density distribution of Figure 1, the density value on the Y axis of flowers’s fig is very low compared to other figures. Also, the numbers on the X axis of flowers's figure are also biased. When this density is expressed as a number of varieties, how many strains does it correspond to? Is it possible to show the raw data for each varieties in addition to other branching characters and flowers? 

As a result of GWAS, genes such as eukaryotic translation initiation factor and myb family transcription factor were listed in the abstract, and these genes were highly correlated with branching traits. However, the authors summarized as Flowing traits due to their high correlation. The correlation between Branching traits and Flowering traits makes a little confusing. The authors evaluates the Flowering Trait, but it would be better to evaluate it as a Branching Traits.

Reviewer 3 Report

Comments and Suggestions for Authors

The objective of the work presented is the identification of genomic regions associated with architectural characters of the cassava plant, which in turn are strongly associated with flowering.
The authors have used a high number of markers (24,040 SNPs used) in a final germplasm panel of 293 varieties. However, the presentation of results and some aspects of their analysis make the work difficult to read and the results quite confusing.

Below I highlight those most relevant aspects:

Title: although the literature supports the high association of branching  with flowering traits in cassava, the title does not seem appropriate to me, since finally the authors analyze only characters related to the architecture of the plant in terms of its height and branching. There is no data that tells us which varieties have flowered, flowering time or nº node of first flower, so I think this should be reflected in the title.

Summary: the summary should include a conclusion of the most important results found, those candidate genes that show a more significant association and their possible role in the associated trait. The term "amount of flowering" seems strange to me, the quantitative way of analyzing it could be by time to (days to) first flower or nº of inflorescences.

Introduction: The sentence (L60) should be changed: "to generate new breeding lines with genetic variabilities which may be beneficial in breeding programs". The term "genetic variabilities is misused. What would be interesting would be to be able to develop, through breeding programs, and thanks to molecular information and greater knowledge of the genetics of the characteristics related to flowering, elite lines of casava that can adapt to the required growing conditions and that present characteristics of uniform flowering and suitable for use.

Also change the phrase (L89) "Although FT orthologs such as MeGI, Cassava GI and CO-like genes (MeCOL1, MeCO, and MeCOL2) have been associated with.."" MeGI and MeCOLs are not orthologs of FT genes. Furthermore, the references given (21,22) study only FT genes.

L98: "It has been used to probe QTL (quantitative trait loci) for developing genetic linkage maps in a number of crops". This phrase does not make sense, linkage maps are those that have been generated to be able to map QTLs.

L100: There is a lot of work on mapping QTLs for flowering, more references of at least species close to casava should really be added.

L106: Incomprehensible phrase: "It is used to mine natural genetic variations 106 in organisms for revelation of variabilities in genetic architectures among traits in crop plants".

In addition, it should be taken into account that when the common name of a crop is mentioned, it must be in lower case, e.g. L116 sesame, L118 soybean.

Materials and methods:
On the one hand, there are paragraphs that are not necessary, such as how the data is taken with a tablet, or how the samples are sent to Australia in plates. However, I miss a better description of the germplasm panel used, for example a supplementary table. In addition, the sowing dates and climatological characteristics of the test areas are also absent. In cassava flowering is stimulated by long days, and  cool temperatures.

Regarding the branching response character, which has been analyzed qualitatively as 0/1, it should not be analyzed as a quantitative character, but rather can be included as another qualitative but morphological marker.

Tables and Figures: In general, I see too many tables and figures that do not contribute much, and yet no supplementary material. Figures 2, 3, 5 could be supplementaries.

Table1: explain the meaning of abbreviations.

Regarding tables 3, 4, 5 and 6, most of the information is repeated. Table 3 could be supplementary. Tables 4 and 5 are exactly the same. Furthermore, in Table 5 it is incomprehensible how if the same SNP is associated with several traits, a single PVE value is given. Is it an average of the characters to which this SNP is associated?. It is also not understood how the SNP S18_1832353 appears twice in the table, but with different genes, two transcripts of the same gene even appear (18G017000.1 and 18G017000.1.2) with different functions, review the table because there are errors.

It would be more informative for the reader to have a supplementary table listing all the genes that are physically found in the selected area for the significant SNPs that explain a variability greater than 5%. And a final table, which includes only the previous genes that, due to their function or the involvement of their ortholog in other species, are selected as candidate genes, providing more information, such as the identification of their orthologs in these other species and the reference of the work of its function.

Another issue that I would like to clarify is that you are talking about a high number of associations but it must be clarified that they are highly correlated traits, therefore we are probably talking about the same associations. In fact, L179 speaks of relatively low correlations with values ​​greater than 0.66, this is not correct.

Results and discussion: These sections need to be rewritten. The results do not contribute anything, only a mere description of tables and figures, while the discussion is too long and its structure seems like results. If the results are sufficiently explanatory and the references are already added, in the end, we are left with a discussion of only the most relevant results and the final conclusions of the work.

All these questions would make the results clearer and could provide an adequate selection of markers and potential candidate genes related to the architecture of the cassava plant.

Comments on the Quality of English Language

It is necessary to check for formatting errors, missing commas and phrases with a meaning that is difficult to understand. I would recommend a final review by a professional.

Round 2

Reviewer 3 Report

Comments and Suggestions for Authors

I believe that although the work provides interesting markers and candidate genes due to their association with the evaluated traits, there are some issues that must be corrected.
The results also need to be improved to be presented more clearly and their discussion more concise, specially the candidate genes part.
Regarding the association analysis, if there are significant differences between localities, it would be interesting to also provide the association analysis separately in each location, to search specific and stable associations.
See my comments in the attached pdf.

Comments on the Quality of English Language

It is ok

Round 3

Reviewer 3 Report

Comments and Suggestions for Authors

The work has gained in its revision and is understood much better and most issues have been clarified and improved. I leave some minor pending issues in the pdf

Comments on the Quality of English Language

The language is much more understandable, only minor formatting issues pending.
